# Examining the Moderating Role of Parental Stress in the Relationship between Parental Beliefs on Corporal Punishment and Its Utilization as a Behavior Correction Strategy among Colombian Parents

**DOI:** 10.3390/children11040384

**Published:** 2024-03-23

**Authors:** Martha Rocío González, Angela Trujillo

**Affiliations:** Facuttad de Psicología y Ciencias del Comportamiento, Universidad de La Sabana, Chía 250005, Colombia; martha.gonzalez@unisabana.edu.co

**Keywords:** corporal punishment, parental beliefs, stress levels

## Abstract

Understanding beliefs about corporal punishment is crucial, as evidence suggests that positive beliefs in its effectiveness predict its use. High parental stress, especially in those valuing corporal punishment, increases the potential for child abuse. Factors such as having many children or low education and socioeconomic status contribute to parental tensions, leading to the use of corporal punishment for behavior correction. We posit that the accumulation of such variables results in heightened stress levels. Our focus aimed to determine the moderating role of stress levels among parental beliefs about corporal punishment and its reported use through quantitative research. In our study, 853 Colombian parents of low, middle, and high socioeconomic status, and from four different regions of Colombia, with children aged 0 to 17 participated. They provided information about their beliefs on corporal punishment, using the Beliefs and Punishment Scale. Correlations indicated that older parents with better socioeconomic status were less inclined to believe that strictness improves children. Regressions suggested that increased belief in corporal punishment modifying behavior, along with higher parental stress, increases corporal punishment use. Moderation models highlighted that when more stressors were present, corporal punishment was used due to stress rather than parental beliefs. Ultimately, stress emerged as a crucial factor influencing corporal punishment use among Colombian parents.

## 1. Introduction

The compelling scientific evidence linking corporal punishment (hereafter CP) to adverse effects on child development is well-established [1,2,3,4,5,6,7]. Despite this knowledge, approximately 1 million children globally endure various forms of violence within their homes [8]. Notably, CP remains a commonly employed disciplinary strategy in numerous countries [9,10].

Corporal Punishment is evidenced by parental actions such as slapping, pinching, kicking, and striking their children with the aim of correcting their behavior. The term “CP” is precisely defined by Straus [7] as the intentional use of physical force to induce pain without causing injuries, with the objective of correcting or controlling a child’s behavior. In the literature, there remains ongoing debate regarding the distinction between corporal punishment and abuse [11], particularly due to the varying degrees of severity associated with corporal punishment. This spectrum ranges from disciplinary actions that are socially accepted in certain contexts and do not result in observable physical harm to the child (e.g., spanking), to actions that are widely regarded as abusive and can cause clear physical harm (e.g., striking with objects). Furthermore, the frequency of corporal punishment varies, occurring either sporadically, such as once a week, or frequently, several days a week. The degree to which corporal punishment is considered normative within a culture may potentially influence its severity.

Research investigating the effects of CP has highlighted that the accumulation of stress during early childhood can influence the structure and function of children’s brains [12,13]. Furthermore, CP has been linked to consequences for cognitive development and academic performance in children [3]. Other studies have associated CP with an elevated likelihood of experiencing partner violence in adulthood [14,15]. Conversely, CP amplifies the risk of intergenerational transmission of violence and serves as a predictor of externalizing behaviors such as aggression, violence, and antisocial behavior conduct [16,17,18]. These behaviors are in turn associated with violence at broader societal levels [19,20,21]. In recent studies, corporal punishment was not associated with any positive developmental outcomes in any culture [22].

Despite the well-documented negative effects found in the literature, CP remains a prevalent disciplinary practice employed by parents across various cultures [9,23]. The question arises: why do parents’ resort to CP in disciplining their children? Scientific evidence indicates that the utilization of CP is influenced by a range of contextual and personal variables. Among these variables, the mental health of the mother has been scrutinized [3,24], along with factors such as gender-based violence, drug use by either parent [25], and a negative perception of the child [26].

Another group of studies has analyzed sociodemographic variables related to children, such as the number of children at home and their age [7,27,28,29,30,31]. Likewise, sociodemographic variables related to parents, such as young age implying less parenting experience [2,32], can contribute to these situations. All these factors, combined with poverty or economic and marital difficulties, may be associated with a lack of emotional regulation and parental stress, which become predisposing factors for the use of CP [2,33]. Parental stress, like general stress, tends to increase the use of CP [27], as well as parents’ beliefs regarding the positive effects or effectiveness of CP on their children [27,34].

Cultural norms, values, and historical contexts are instrumental in shaping attitudes towards corporal punishment. While certain societies prioritize obedience and authority [35,36], resulting in its widespread acceptance, others emphasize non-violent forms of discipline [37]. For instance, collectivist cultures may prioritize group harmony and respect for authority, leading to a more frequent use of corporal punishment. Conversely, individualist cultures underscore individual rights and autonomy, favoring non-violent disciplinary methods [38]. Moreover, family structures, dynamics, and socio-economic factors contribute significantly to the cultural variation in the use of corporal punishment. Ultimately, the relationship between caregivers and children is profoundly intertwined with cultural expectations and norms, thereby influencing the disciplinary methods employed and their perceived effectiveness [39].

Extensive data on the global prevalence of CP exists [9,10,23], along with scientific evidence elucidating the correlation between its utilization and parental beliefs used to justify it [2,23,40,41]. While some studies have explored the connection between parenting stress and CP [42,43,44,45,46,47,48,49], we have investigated the moderating role of stress between the prevalence of CP and the beliefs justifying its use. Thus, the objective of this study was to assess the moderating role of stress in the relationship between parental beliefs about CP and their reported use of it. Grounded in existing theory, the expectation was that prevalence would exhibit a positive association with beliefs about its use. Moreover, it was anticipated that in the presence of higher levels of stressful factors, CP would be employed due to stress rather than parental beliefs.

### 1.1. Prevalence of CP and Beliefs Justifying Its Use

Several studies have demonstrated the widespread occurrence of CP against children within households, frequently surpassing a prevalence rate of 50% in international samples [7,50,51]. This establishes CP as a substantial public health concern, as acknowledged by the World Health Organization in 2022 [52]. Despite its global significance, there exists a notable dearth of information concerning the prevalence and patterns of CP specifically in the context of Latin America [9].

The findings from the study conducted by Trujillo et al. [33] indicate that 77% of Colombian parents utilized CP at least once in the past year. A comparative analysis with data from other countries in the Latin American region underscores the high prevalence of CP in the Colombian population. For instance, in Chile, 48% of 1151 participating men reported employing CP at home [53]. UNICEF [10] reports that in Argentina, statistical data reveal that 72% of children aged 2 to 14 experienced CP at home. In Brazil, among 744 participating men, 36% acknowledged having used CP on their children during childhood. In Bolivia, 48.7% of women reported instances of CP towards children in their homes. Additionally, 44% of children in Ecuador were subjected to CP [53]. In a 2008 study in Bolivia involving 10,092 women aged 15–49 with children, 48.7% reported incidents of CP towards children in their homes [10].

The elevated prevalence of CP in Colombia, compared to other countries, may potentially be attributed to the nearly 50-year armed conflict, which could normalize negative interactions such as CP. Studies conducted in nations with a protracted history of armed conflict have demonstrated that patterns of violence permeate the culture, manifesting in various forms, including sexual violence, partner violence, and physical and emotional abuse towards children [54]. In this context, findings from a study conducted among the Colombian population suggest that psychosocial stressors, both within the community and family, play a role in mothers’ utilization of physical punishment in Colombia. Particularly, mothers’ past experiences of corporal punishment by their own parents significantly heightened the probability of employing corporal punishment with their young children [9]. Several studies indicate that parents who have experienced forms of child abuse are at a higher risk of perpetrating abuse against their own children [55].

On the other hand, attitudes toward domestic violence, homicide rates within the municipality, and levels of poverty at both the family and neighborhood levels emerged as significant predictive factors for mothers resorting to hitting their young children with objects [9].

In essence, the impact of war can legitimize violence, creating a context in which parents may perceive justification in resorting to punitive measures against their children [56,57]. Parental beliefs, as defined by the Government of Colombia [58] (p. 30), encompass explanations, thoughts, and principles regarding methods of educating, guiding, and disciplining children. Previous research establishes a predictive relationship between these beliefs and the use of CP [59], emphasizing their role in shaping the practices parents employ to correct their children’s behavior [60]. Moreover, studies suggest a positive association between the use of CP and the potential for physical abuse, particularly among parents harboring high levels of favorable beliefs about the use of this parenting practice [2,27,41,50,61].

Moreover, studies across various cultural groups reveal that the primary factor influencing parental approval of CP is the belief in its normativity, considering it an essential aspect of parenting, even for infants [23]. A comprehensive literature review conducted by Chiocca [62] identifies one of the key factors influencing the utilization of CP: parents’ belief in its normative nature, considering it as the appropriate approach to child rearing. The review indicates that such beliefs, coupled with specific stressors encompassing parent–child interactions and external situations, serve as triggers for the use of CP.

### 1.2. Parenting Stress and CP

Parenting stress is defined as a psychological mechanism that induces distress and frustration in parents concerning activities specifically associated with raising their children [63]. Consequently, when parents experience frustration in their parenting roles, it often triggers emotions such as anger. These emotions, coupled with impulsivity and a lack of emotional regulation, can pose challenges for parents in maintaining control over their behavior, potentially leading to the adoption of negative parenting practices, such as CP [2,44].

Parenting stress serves as a predictor for the implementation of severe disciplinary measures [64]. Numerous studies have illustrated a direct relationship between parenting stress and the use of CP, suggesting that the predictive variables of CP are, in turn, factors contributing to increased parenting stress [42,44,45,46,47,48,49]. Consequently, heightened parental stress is associated with an increased tendency to resort to severe disciplinary methods, including CP [65,66,67,68].

Certain sociodemographic variables have been linked to parenting stress and the use of CP. In terms of the number of children, multiple studies have demonstrated an association between an increased number of children in a family and a higher likelihood of employing CP [7,27,29,31]. Research suggests that circumstances such as parents having limited time to fulfill their parenting responsibilities for each child contribute to heightened stress, prompting the use of CP as a swift method of behavior control [29,34]. Another related factor involves elevated economic demands, leading to longer working hours for parents and, subsequently, reduced time available for children’s education [7].

Regarding the age of children, previous research suggests that CP tends to increase from infancy to the age of 2, remains relatively stable during the ages of 3 to 5, and then steadily decreases from ages 5 to 17 [2,69]. Gershoff’s [2] review study corroborated these findings and highlighted that parents often perceive CP as more suitable for preschool-aged children, while considering it less appropriate for infants and children aged 5 and older. Furthermore, she observed a significant decline in the rates of CP as children progress into adolescence.

Another influential variable related to parenting stress and CP is the age of the parents. Studies indicate that younger parents are more inclined to use CP [2,32]. This tendency could be attributed to higher levels of impulsivity and parental stress associated with limited parenting experience among younger parents [2].

The literature consistently shows that factors such as having many children, low levels of education, or a lower socioeconomic status contribute to increased tensions in parents, ultimately leading to the utilization of CP as a method for correcting children’s behavior [33,70]. Given the demonstrated impact of these variables or situations on heightened stress levels, our focus was on exploring the moderating role of stress among various beliefs that parents uphold regarding CP and their reported use of it. This investigation aimed to understand how stress levels may influence the relationship between parental beliefs and the actual application of CP.

**Hypothesis** **H1:**
*Considering the above review of the literature, our hypothesis is that stress moderates the relationship between beliefs in corporal punishment and its use, such that under high levels of stress, individuals with stronger beliefs in corporal punishment are more likely to resort to its use as a disciplinary measure compared to those with weaker beliefs.*


## 2. Materials and Methods

### 2.1. Participants

The participants were 853 parents (mean age 35.06, SD: 8.25) of children between 0 and 17 years old. The sample was selected to be representative of the large, urban population of the four main cities of Colombia, from which it was drawn (Medellín: n = 143; Barranquilla: n = 127; Bogotá: n = 463; Cali: n = 120). All 853 parents reported information about their beliefs toward CP.

The following table shows the sociodemographic characteristics of the sample (Table 1).

### 2.2. Measurements

Each parent (mother or father) completed a consent form, a demographic questionnaire in addition to the Beliefs and Punishment Scale [71]. This scale has six questions in which parents express their agreement or disagreement with the use of punishment as a way of correcting children’s behavior. They can also mark an option of “I don’t Know” or an option of “I prefer not to answer”. Once respondents provided their ratings for each item, scores on the scale were calculated by summing or averaging the responses across all items. This yields an overall score that reflects the respondent’s attitudes or beliefs regarding corporal punishment.

Parents also answered the scale of physical assault of the Spanish version of the Parent–child Conflict Tactics Scale (CTSPC) [72]. The questionnaire measures three dimensions: Non-violent Discipline, Psychological Assault, and Physical Assault. For the present study, we used the Physical Assault scale, which measures a variety of acts ranging from CP that is socially legitimized, to criminal acts of physical assault. It also has a series of items that measure the frequency in which these punishments are used in the last year. Participants respond to each item using an 8-point scale from 1 = Once in the past year, to 6 = More than 20 times in the past year. It also has tow options that are: 7 = Not in the past year, but it happened before, and 0 = This has never happened. The scale is scored by adding the midpoints for the response categories chosen by the participant. Higher total scores indicate greater endorsement of behaviors related to parent–child conflict. The scale provides insights into the frequency and severity of different conflict tactics used by parents.

### 2.3. Procedure

Participants were recruited through a snowball sampling technique where those who participated gave us the contact information of other subjects who could be interested in participating in the research. We use telephone and an online survey for participants to answer. For the telephone survey, two psychologists previously trained, called people, explained the objective of the research, and asked them if they wanted to participate. If the person agreed, the researchers read the informed consent, and once the subject agreed to it, they applied the survey. The telephone survey usually lasted around 10 min, and at the end, the subject was free to give us contact information about other subjects that could be interested in participating.

The online survey format Is the same as the telephone survey, but with written instructions and informed consent. The questionnaire was uploaded using Google Forms. A link was sent through private groups on social networks explaining that it was a questionnaire for parents of underage children that lived in one of the four cities of the study.

The study was approved by the ethical review board of the Psychology Department of La Sabana University (Minute 083, 13 May 2015).

## 3. Results

To cluster the assessed beliefs, considering potential high correlations among them, a factorial analysis was conducted using principal axis analysis and the Oblimin rotation method. The Kaiser–Meyer–Olkin (KMO) measure of sample data adequacy was 0.77, indicating the suitability of the factorial analysis, based on the classification proposed by Kaiser and Rice [73]. Additionally, Bartlett’s test of sphericity was significant (*p* < 0.001), suggesting a high likelihood of data relatedness [74]. The highest observed inter-item correlation coefficient was 0.76. Consequently, two dimensions were derived, explaining 77.6% of the variance. The first factor comprised three beliefs associated with the use of CP to modify child behavior, with loads ranging from 0.76 to 0.75. Two beliefs that did not exhibit significant correlation with each other were retained as individual variables: “If CP was good for me, it is good for my children” and “The stricter the parents, the better the children”.

Given the consistent association of sociodemographic variables with CP in prior studies, our initial purpose was to examine associations between the three beliefs about CP and various sociodemographic variables. This exploration aimed to ascertain whether these beliefs are also correlated with key parental factors. Pearson correlations were conducted, considering associations that demonstrated a significant level of 0.001 (two-tailed). The variables examined encompassed the age of the parents (we use the continuous variable of age, not the grouped one), educational level (we used a scale from 1 = primary, to 5 = postgraduate), and socioeconomic status (been 1 low, and 3 high status). As indicated in Table 2, the correlations suggest that the younger the parents, with lower socioeconomic status and educational levels, are more inclined to believe that being strict leads to better outcomes for children. Conversely, educational level and socioeconomic status exhibit a positive correlation with the belief that CP worked for them, and therefore, it should work for their children. In contrast, these same two variables demonstrate a negative correlation with the belief that CP will modify the child’s behavior. Notably, no significant association was identified with the gender of the parent.

To assess whether the use of CP could be explained by parental beliefs and the level of stress, a logistic regression was conducted, introducing the three beliefs and the stress index. As depicted in Table 3, it was found that the more parents believe that CP modifies behavior or that being stricter improves the child, the more they engage in this practice. Additionally, it was identified that higher levels of stress in parents are associated with increased use of CP.

To explore the moderating influence of stress levels on the three parental beliefs about CP and its reported usage, we initially employed a single moderation regression model for the anticipated moderator. This analysis utilized the PROCESS macro v. 16.3 for SPSS 26 [75], incorporating bootstrapped standard errors (5000) and mean-centered products. The results showed a positive correlation between the three types of beliefs and the use of CP. However, the moderation model suggested that the impact of believing CP modifies a child’s behavior is diminished when the level of stress is high (refer to Table 4, and Figure 1). Additionally, the effect of believing that if CP worked for the parents, it should work for their child, is also attenuated when the level of stress is elevated (see Table 4). On the contrary, concerning the parental belief that stricter parenting leads to better child outcomes, our findings indicate that the relationship with CP is not influenced by stress levels (see Table 4), but in fact, for the three beliefs, the interaction with stress is.

Examining the moderation probes, the disparity arises when parents believe that CP should be applied to their children because it proved effective for them.

In summary, our findings indicate that in the presence of heightened stressors, the use of CP tends to be a response to stress rather than a result of the parent’s personal beliefs. Specifically, this pattern is observed concerning the belief that CP modifies behavior and the belief that being stricter leads to better outcomes for their children. However, it was noted that the belief that if CP worked for the parents, it should work for their children did not exhibit moderating effects with stress.

## 4. Discussion

The aim of this study was to determine the moderating role of stress among parental beliefs regarding CP and its reported use. The findings of the study contribute evidence regarding the significance of beliefs [2,27,41,50,61] and high levels of stress [42,44,45,46,47,48,49] in the utilization of CP.

Furthermore, the results enhance understanding of predictors of CP, suggesting that when parents experience high levels of stress, the likelihood of employing CP increases [65,66,67,68]. Specifically, having a high index of stressors, including having more than three children at home [7,27,29,31], being a young parent [2,30,32], and having a low socioeconomic status, is associated with the use of CP [33].

On the other hand, the results demonstrate an association between younger parental age and the belief that their children will be better if they are stricter. This association likely occurs because young parents have less experience and are less prepared for parenthood [76]. Additionally, young parents believe that authority is associated with more punitive practices like CP, and that being strict is not sufficient for their upbringing.

Similarly, the results indicate that lower levels of education and socioeconomic status [30] are associated with two beliefs: CP modifies the child’s behavior, and the stricter the parents, the better the children. This finding contrasts with the study by Qasem et al. [77], in which lower educational levels were positively associated with the acceptance of CP.

In contrast, the belief that if CP worked for me, it would work for my child, is associated with higher levels of education and socioeconomic status. Regarding this, Weller et al. [78] point out that personal history of interaction with events or situations is an important influence on the development of disciplinary beliefs. Individuals who were punished tend to believe in the appropriateness of punishment as a parenting practice, regardless of their socioeconomic and educational levels.

The moderation analysis revealed that the impact of believing that CP modifies children’s behavior and that if CP worked for me, it would work for my child, diminishes when the level of stress is high. In this regard, the study by Yoon et al. [79] demonstrated a significant positive relationship between parental stress and the risk of child maltreatment, even after controlling for other predictors. Furthermore, no moderating effect of social support was found to buffer the negative impacts of parental stress on the risk of child maltreatment. This finding suggests that parental stress directly affects the use of CP and other punitive practices, even in the presence of acceptance beliefs, thereby confirming the initial hypothesis of the study, which expected prevalence to be positively associated with beliefs about its use, and that higher levels of stressful factors would lead to greater use of CP by parents.

In contrast, and regarding the belief that the stricter the parents, the better the children, no influence of the level of stress was found. This result may be related to the concept of being strict, which is not the same as using CP to change children’s behavior, or because if CP worked for me, it would work for my child.

Overall, the results indicate that beliefs justifying CP are present across all educational and socioeconomic levels, contrasting with the study by Trujillo et al. [33] in the Colombian population, where no differences were found in CP utilization based on socioeconomic and educational levels. This may be because a culture permeated by violence, as is the case in Colombia, exerts a powerful influence on the development of beliefs, which in turn could lead to violent behaviors within the family and society [80]. However, no study has been conducted to assess this issue in the Colombian population.

Furthermore, the general results demonstrate that parental stress is a significant psychological mechanism influencing the use of CP. According to Khalifa [29], CP could be used by parents from all social, economic, and educational backgrounds to discipline their children due to increased stress levels leading to reduced parental tolerance. This may occur because when a parent attempts to control their child’s behavior through CP, it could be due to the stress of parenting or difficulties in regulating their emotions and a limited repertoire of positive parenting practices to educate their children, all within a culture of violence that normalizes CP as a form of discipline [2].

Indeed, different authors suggest that stressful life events can result in an inability to respond to situations appropriately, partly due to poorly regulated emotions [81]. Interpersonal trauma, including interpersonal violence, compared to other types of stressful life events, could have a more profound effect on emotion regulation processes [82]. The hypothesis presented here is that parental CP in Colombia may be related to their lack of emotional control, perhaps arising from their direct or indirect exposure to war situations.

Although parents may not be aware of the normalization of a culture of violence in everyday life, as expressed above, or their difficulty in regulating emotions, the truth is that CP is used more frequently when parents are angry [83,84,85] or when they experience episodes of frustration in their daily lives [86,87]. Cultural determinants exert a substantial influence on the acceptance and prevalence of corporal punishment, as well as on the dynamics within the caregiver-child relationship.

Also, the emotions experienced by parents during interactions with their children influence how they perceive and, in turn, react to children’s inappropriate behaviors [85,88]; if their emotional response is too strong, parents are less capable of regulating their emotions and, consequently, their behavior [88,89]. Therefore, when parents are upset or emotional, they tend to make negative attributions about their children’s behaviors and to resort to power assertion, such as CP, as a response [85,86,90]. This is why Gelles and Straus [91] propose the elimination of norms and cultural values that accept violence to resolve conflicts and problems within the family. Similarly, it is evident that culturally more competent interventions should be provided, aiming to reduce parental stress rather than solely modifying beliefs.

### 4.1. Practical Implications of the Research

Different countries have incorporated educational interventions related to CP into their national parenting programs [92,93]. According to the results of this study, first, it is necessary for positive parenting practices programs to reframe the normalization of violence. This implies understanding the negative effects that CP can have, its implications on children’s rights, even when it is not severe [94]. Parents, especially in Colombia, must understand that CP is a manifestation of a culture of violence that needs to change in order to create a culture of peace in the country.

The findings of this research have several practical implications for interventions and policies aimed at reducing the use of corporal punishment among Colombian parents. The results underscore the necessity of developing programs tailored to effective stress management for parents [95]. Mitigating maladaptive reactions to stress necessitates addressing negative perceptions about child behavior [96] and diminishing parental anger toward normative child behaviors [97]. Initiatives aimed at fostering positive parenting practices should prioritize strategies to mitigate parental impulsivity and enhance emotional regulation, particularly in managing anger, thereby reducing reliance on CP. Recognizing the association between parental stress levels and the use of corporal punishment underscores the importance of addressing stress management strategies as part of parenting support programs. Providing resources and interventions that help parents cope with stress effectively can potentially reduce reliance on corporal punishment as a disciplinary measure.

Secondly, understanding the moderating role of stress levels in the relationship between parental beliefs about corporal punishment and its reported use highlights the need for targeted interventions. Programs designed to address parental beliefs about discipline should also incorporate components that specifically target stress reduction techniques. By addressing both factors simultaneously, interventions may be more effective in promoting positive and non-violent parenting practices.

Moreover, the identification of differences in beliefs about corporal punishment among parents of varying socioeconomic statuses and ages suggests the importance of tailoring interventions to specific demographic groups. For instance, interventions targeting older parents or those with lower socioeconomic status may need to address unique cultural or contextual factors that influence their beliefs about discipline and parenting practices.

Overall, the research emphasizes the importance of comprehensive and culturally sensitive interventions that address parental stress, beliefs about corporal punishment, and contextual factors to promote positive parenting practices and reduce the use of corporal punishment among Colombian parents.

### 4.2. Limitations of the Study and Future Research

There are several limitations to our study that should be consider: The study focused on Colombian parents from various socioeconomic backgrounds and regions. However, the sample may not fully represent the entire Colombian population. Also, generalizability to other cultural contexts or diverse populations might be limited.

The study’s cross-sectional design limits causal inferences. Longitudinal data would enhance understanding of stress-corporal punishment dynamics. Also, social desirability bias might influence responses, affecting the accuracy of reported corporal punishment practices. Also, focusing solely on the belief that strictness improves children overlooks other multifaceted parental beliefs.

In summary, stress has surfaced as a pivotal determinant influencing the utilization of corporal punishment among Colombian parents. To advance our understanding and facilitate the development of effective interventions and policies, future research must tackle the identified limitations. Particularly, investigating age-specific analyses of parenting practices could provide valuable insights, given the variability observed across different developmental stages. Furthermore, while moderation models were explored, the examination of mediation effects remains unexplored, leaving underlying mechanisms uncharted. Exploring these mediation pathways could offer illuminating perspectives on the intricate dynamics underlying the use of corporal punishment.

## 5. Conclusions

It is concluded that both stress indicators and beliefs are relevant variables for understanding the use of CP. Additionally, a close relationship between them is observed: as stress levels increase, some beliefs are affected. However, the results suggest that the level of stress has a more significant impact on the use of CP than the beliefs themselves.

Furthermore, it is concluded that a culture of violence and other sociodemographic variables may be linked to lack of emotional regulation and parental stress, which in turn predisposes to the use of CP. This use of CP is associated with aggressive, violent, and antisocial behaviors in children, contributing to increasing levels of violence in society. In other words, violence and the use of CP reinforce each other in a cycle that can further increase violence.

## Figures and Tables

**Figure 1 children-11-00384-f001:**
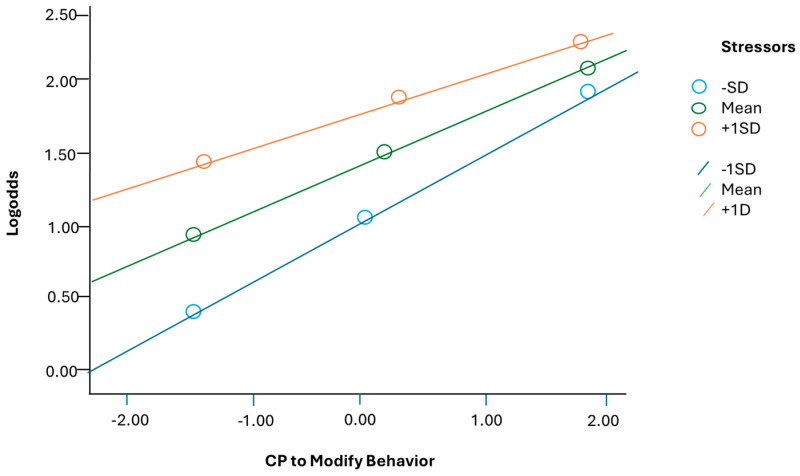
Moderation effects of stress on the probability to use CP when parents believes that CP modify behavior.

**Table 1 children-11-00384-t001:** Sociodemographic characteristics of the parents’ sample.

Variables	Bogotá	Cali	Medellin	Barranquilla	TOTAL
	%	%	%	%	%
**Parents**					
Mothers	88.3	90.8	89.5	93.7	90.6
Fathers	11.7	9.2	10.5	6.3	9.4
**Age range**					
15 to 20	4.7	1.6	3.4	31.3	10.25
21 to 30	21.1	18.5	36.7	47.3	30.9
31 to 40	49	54	43.5	19.1	41.4
41 to 50	21.1	25	11.6	1.5	14.8
51 to 60	3.8	0.8	4.8	0.8	2.5
61 to 70	0.4	0	0	0	0.1
**Education**					
Primary	7.8	9.2	14	5.5	9.1
High school	38.8	35.8	29.4	23.6	31.9
Technical	19.3	17.5	17.5	29.1	20.8
Professional	18.8	18.3	19.6	33.1	22.4
Postgraduate	15.2	19.2	18.9	8.7	15.5
**Socio-economic status**					
Low (Classes 0, 1 and 2)	40.8	48.4	37.4	45.8	43
Middle (Classes 3 and 4)	39.9	32.3	34	30.5	34.17
High (Classes 5 and 6)	19.3	19.4	28.6	23.7	22.7
**Occupation**					
Employee	39.7	53.2	41.5	30.5	41.22
Independent	26.6	24.2	27.2	35.1	28.2
Unemployed	5.8	2.4	4.8	4.6	4.4
Stay-at-home parent	27.9	20.2	26.5	29.8	26.1

**Table 2 children-11-00384-t002:** Pearson correlation between parent’s beliefs about CP and sociodemographic variables.

	CP to Modify Behavior	If CP Worked for Me, It Should Work for My Child	The Stricter the Parents, the Better the Child
Gender	0.048	−0.033	0.061
Age	−0.036	0.039	−0.096 **
Educative level	−0.115 **	0.086 *	−0.246 ***
SES	−0.140 ***	0.135 ***	−0.288 ***

* *p* < 0.05; ** *p* < 0.001; *** *p* < 0.0001.

**Table 3 children-11-00384-t003:** Logistic Regression for Parental beliefs and stress level.

	B	S.E.	Wald	df	Sig.	Exp(B)
CP to modify behavior	0.220	0.081	7.443	1	0.006	1.246
CP worked for me it should work for my child	0.108	0.046	5.487	1	0.019	1.114
The stricter the parents, the better the child	0.112	0.032	12.494	1	0.000	1.118
Level of Stress	0.362	0.119	9.268	1	0.002	1.437

**Table 4 children-11-00384-t004:** Conditional linear regressions of parental beliefs on CP use with moderation of stress level.

“CP to modify behavior”				
	Coeff.	SE	Z	*p*
Constant	1.37	0.09	14.73	0.00
Stress level	0.42	0.11	3.69	0.00
CP to modify Behavior	0.37	0.07	5.14	0.00
CP to modify behavior × Stress level	−0.14	0.08	−1.79	0.07
Stress level conditional effects	Coeff.	SE	Z	*p*
Low Stress level	0.47	0.09	5.19	0.00
Mean Stress level	0.37	0.07	5.14	0.00
High stress level	0.25	0.10	2.50	0.01
“If it worked for me, it should work for my child”				
	Coeff.	SE	Z	*p*
Constant	1.32	0.09	14.86	0.00
Stress level	0.36	0.11	3.28	0.00
It worked for me it works for my child	0.17	0.04	3.94	0.00
If it worked for me...× Stress level	−0.09	0.05	−1.85	0.06
Stress level conditional effects	Coeff.	SE	Z	*p*
Low Stress level	0.24	0.06	4.09	0.00
Mean Stress level	0.17	0.04	3.94	0.00
High stress level	0.09	0.06	1.49	0.14
“the stricter the parents the better the child”				
	Coeff.	SE	Z	*p*
Constant	1.35	0.09	14.30	0.00
Stress level	0.52	0.12	4.27	0.00
The stricter the parents...	0.16	0.03	5.17	0.00
The stricter the parents...× Stress level	0.00	0.04	0.07	0.94
Stress level conditional effects				
	Coeff.	SE	Z	*p*
Low Stress level	0.16	0.04	4.55	0.00
Mean Stress level	0.16	0.03	5.17	0.00
High stress level	0.16	0.06	2.97	0.00

## Data Availability

Publicly available datasets were analyzed in this study. This data can be found here: [https://drive.google.com/drive/folders/1iIgDYEHb-E6LqIr8FqaeNL_Lr2kfTYnl?usp=sharing].

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
