# Peer review of "Examining the Moderating Role of Parental Stress in the Relationship between Parental Beliefs on Corporal Punishment and Its Utilization as a Behavior Correction Strategy among Colombian Parents"

_children, 2024, doi:10.3390/children11040384_

Round 1
Reviewer 1 Report
Comments and Suggestions for Authors
Dear Authors,
I appreciated reading your paper, and I only have two minor suggestions:
1) Please, specify in the abstract already the meaning of CP and PP (if not in the title). I assume this can help readers who are screening publications to consult since not everybody, especially non-native speakers, may be familiar with acronyms.
2) I would probably emphasize more that this paper takes a strong individualistic approach. Even though culture is mentioned in the body of the text, to me the real focus here seems to be on the micro-level (more than the meso- or macro-level). Which is fine; I just believe it could be further specified.
Wish the authors good luck with their endeavor.
Author Response
Thank you very much for taking the time to review this manuscript. Please find the detailed responses in the attached word and the corresponding corrections highlighted changes in the re-submitted files

Reviewer 2 Report
Comments and Suggestions for Authors
The article presents a subject of notable interest, such as corporal punishment. It makes a theoretical positioning that adequately establishes the objective of the research, where the term and other related variables are discussed. However, a small section is missing in this theoretical review of what corporal punishment is, as well as the different forms of manifestation. It must be taken into account that when physical punishment is exceeded in its forms and frequency, it can lead to child abuse, even if this is not really the paternal intention.
On the other hand, the table of participants is very complete, but in the analysis carried out the variables used are not known. For example, we see that the age of the parents ranges between 15 and 70 years. It is necessary to provide mean and standard deviation data. But in the results they refer to the youngest parents. What age are the authors referring to? to the group of 15 to 20 years old? , or some else?. It is not known what age variables the data analyzes have collected. The same thing happens with the educational novel. That is, in both factors there are many levels, which need to be specified in the results, since they have referred to as sociodemographic variables to consider.
Finally, although they carry out a good bibliographic review, it would be necessary, given the topicality of the topic, to update this review, since the percentage of bibliographic references of the last 6 years (since 2018) does not reach 27%.
Author Response
Thank you very much for taking the time to review this manuscript. Please find the detailed responses in the attached word and the corresponding revisions/corrections highlighted changes in the re-submitted files

Reviewer 3 Report
Comments and Suggestions for Authors
The work the authors propose seems very interesting and contributes to extending our current knowledge on the phenomenon of child maltreatment. I think the contribution is correctly placed in the special issue and can make a stimulating contribution.
I ask the authors to consider taking action on these points:
- do not include acronyms or abbreviations in the title and abstract. Insert abbreviations alongside the full word in the text at the first opportunity.
- I believe that the topic of corporal punishment is also culturally determined and that the relationship with caragivers (parents but also teachers) is culturally based. This aspect should at least be disussed, albeit secondary to your objectives, as it is not a cross-cultural study
-Please create a section on hypotheses and express them clearly.
- The abstract should be more informative, state the type of study (research design), the cultural context in which it was conducted, and the socio-demographic characteristics of the ampion.
- In my opinion, it is clearer if you report the average age of the total sample and the gender prevalence.
- please report information on ethics committee approval.
- Please better describe the limitations of the research and future implications.
- Please extend (by creating a separate section) the practical implications of the research.
- Could it be useful to briefly mention the topic of intergenerational transmission of trauma? (Example: https://doi.org/10.1007/s40653-019-00273-1)
- Finally, I would like to ask the authors to further clarify how the scales of the instruments used were calculated.
Author Response
Thank you very much for taking the time to review this manuscript. Please find the detailed responses on the attached document and the corresponding revisions/corrections highlighted/in track changes in the re-submitted files.

Round 2
Reviewer 2 Report
Comments and Suggestions for Authors
The authors have taken the suggestions into account, and although they have not been able to do them all, because I understand that they should remake the article, they have made a substantial improvement so that it can be accepted for publication in the journal.
Reviewer 3 Report
Comments and Suggestions for Authors.